# Changes in the Epidemiology of Diabetic Retinopathy in Spain: A Systematic Review and Meta-Analysis

**DOI:** 10.3390/healthcare10071318

**Published:** 2022-07-16

**Authors:** Pedro Romero-Aroca, Maribel López-Galvez, Maria Asuncion Martinez-Brocca, Alicia Pareja-Ríos, Sara Artola, Josep Franch-Nadal, Joan Fernandez-Ballart, José Andonegui, Marc Baget-Bernaldiz

**Affiliations:** 1Ophthalmic Department, University Hospital Sant Joan, Institut de Investigacio Sanitaries Pere Virgili, Universitat Rovira & Virgili, 43204 Reus, Spain; mbaget@gmail.com; 2Ophthalmic Department, IOBA, Hospital Clínico Universitario de Valladolid, 47003 Valladolid, Spain; maribel@ioba.med.uva.es; 3Comprehensive Healthcare Plan for Diabetes, Regional Ministry of Health and Families of Andalusia, Government of Andalusia, 14071 Seville, Spain; masuncion.martinez.sspa@juntadeandalucia.es; 4Ophthalmic Department, Hospital Universitario de Canarias, Carretera Ofra S/N, 38320 San Cristóbal de La Laguna, Spain; aparejar@gmail.com; 5Family Physiacian, Health Care Center Jose Marva, 28020 Madrid, Spain; sara.artola@gmail.com; 6EAP Raval Sud—Gerencia Territorial Barcelona, Institut Catala de la Salut, IDIAP Jordi Gol/CIBERDEM (IIB Sant Pau), 08006 Barcelona, Spain; josep.franch@gmail.com; 7Area of Preventive Medicine and Public Health, Faculty of Medicine and Health Sciences, Institut d’Investigacions Sanitàries Pere i Virgili, Universitat Rovira i Virgili (URV), 43003 Tarragona, Spain; joan.fernandez-ballart@urv.cat; 8CIBER Fisiopatología de la Obesidad y Nutrición (CIBERobn) Instituto Carlos III, 28029 Madrid, Spain; 9Department of Ophthalmology, Hospital Universitario de Navarra, IdiSNA, Navarra Institute for Health Research, 31009 Pamplona, Spain; joseandonegui@gmail.com

**Keywords:** diabetic retinopathy prevalence, diabetic retinopathy incidence, epidemiology, diabetes mellitus, screening, systematic review, meta-analysis

## Abstract

Background. The aim of the present study was to determine the prevalence and incidence of diabetic retinopathy (DR) and its changes in the last 20 years in type 2 diabetes mellitus (T2DM) patients in Spain. Methods. A systematic review with a meta-analysis was carried out on the studies published between 2001–2020 on the prevalence and incidence of DR and sight-threatening diabetic retinopathy (STDR) in Spain. The articles included were selected from four databases and publications of the Spanish Ministry of Health and Regional Health Care System (RHCS). The meta-analysis to determine heterogeneity and bias between studies was carried out with the MetaXL 4.0. Results. Since 2001, we have observed an increase in the detection of patients with DM, and at the same time, screening programs for RD have been launched; thus, we can deduce that the increase in the detection of patients with DM, many of them in the initial phases, far exceeds the increased detection of patients with DR. The prevalence of DR was higher between 2001 and 2008 with values of 28.85%. These values decreased over the following period between 2009 and 2020 with a mean of 15.28%. Similarly the STDR prevalence decrease from 3.67% to 1.92% after 2008. The analysis of the longitudinal studies determined that the annual DR incidence was 3.83%, and the STDR annual incidence was 0.41%. Conclusion. In Spain, for T2DM, the current prevalence of DR is 15.28% and 1.92% forSTDR. The annual incidence of DR is 3.83% and is 0.41% for STDR.

## 1. Introduction

Diabetes mellitus (DM) is a chronic disease that affects 537 million adults worldwide; this number is projected to rise to 643 million in 2030 and 783 million in 2045. In Europe, 1 in 11 adults are living with diabetes, and this is expected to reach 67 million by 2030 and 69 million by 2045 [1]. More than one in three (36%) adults living with diabetes are thought to go undiagnosed. Data collected from the diabetes study [2,3] published in Spain in 2012 evaluated the prevalence of diabetes mellitus of around 14% of the adult population, with type 2 diabetes (T2DM) being the most prevalent, representing 90% of cases. This equates to 11.6 cases per 1000 people/year, and the forecast is that by 2030, one in ten adults will have diabetes [1].

Diabetic retinopathy (DR), a complication of diabetes, is one of the main causes of blindness in young adults worldwide [4,5]. The duration of the disease, the type of diabetes treatment, and the degree of metabolic control are all determining factors for its development.

DR screening is cost effective and is essential if we are to prevent an increase in cases of poor vision and blindness in the diabetes population [6]. Screening is carried out through non-mydriatic retinography, and the various scientific societies recommend it is repeated annually or at least biannually [7,8].

The epidemiology of DR in Spain has been changing as its screening programs have been implemented. The DR-screening programs were implemented by the Spanish Ministry of Health (MoH), and a diabetes strategy was published for the National Health System (RHCS) both recommended early diagnosis, screening, monitoring, treatment, and adequate control of chronic complications of DM [9,10].

The objective of the present study is to carry out a study on the DR epidemiology in Spain, determining the current prevalence and incidence and describing the changes that have occurred in the last 20 years after the implementation of screening programs through a systematic review and meta-analysis of published studies.

## 2. Material and Methods

Setting and design: This was a systematic review study with the meta-analysis of publications from 2001 to 2020 on the prevalence of and incidence of DR in T2DM patients in Spain after implementing the screening systems.

### 2.1. Literature Research and Study Selection

We conducted a systematic review of the literature to identify all relevant publications on the prevalence and incidence of DR in Spain. We followed the guidelines of preferred reporting items for systematic reviews and meta-analyses (PRISMA) [11] and the international guidelines for systematic literature reviews and meta-analyses of observational studies and epidemiology (MOOSE) [12].

Database. A general data search included the following databases, EMBASE, Web of Science, Scopus, and MEDLINE, complemented by a Google Scholar search. The search was made for Spanish studies that have been published since 1 January 2001 to 31 December 2020. Publications in English and Spanish were searched using the keywords: “diabetes mellitus, diabetic retinopathy, screening, prevalence, and incidence,” their equivalent descriptions in Spanish, and all the different combinations of those keywords to find the maximum possible number of citations (Appendix A).

### 2.2. Inclusion Criteria

Studies of patients with T2DM in Spain published from 1 January 2001 to 31 December 2020;Population-based studies, cross-sectional or longitudinal type;Studies on screening of DR in Spain; Studies that have provided data on the incidence or prevalence of DR in Spain;The studies must have provided a clear definition of DM made by general practitioner or endocrinologist;The studies must have provided a clear definition of DR.

### 2.3. Exclusion Criteria

Excluded were studies published after the year 2001 but carried out before 31 December 2000; those defined as clinical series or those studies based only on patients treated in hospitals; duplications of the same study in different journals; studies in which the full text were not available; and studies published in the form of an abstract or summary as a result of presentations at conferences.

### 2.4. Data Extraction

After the initial database search, abstracts were screened, and if eligible, full texts underwent further assessment for eligibility by two authors reassessing all full-text articles. Any ambiguity or disagreement between the authors was resolved by discussion between them. The full texts of potentially relevant publications were retrieved. If more information was needed, the authors of the publications were contacted.

### 2.5. Quality Assessment

To assess the risk of bias and quality of the studies included in this systematic review, the full texts of eligible publications were screened using a checklist. The checklist was based on the STROBE [13] (strengthening observational study reporting in epidemiology to evaluate observational studies) principles for primary observational studies and MOOSE [11].

### 2.6. Diabetic Retinopathy Definition and Assessment

The definition of DR in the studies was accepted if it was in accordance with the early treatment of DR study (ETDRS) [14] or the international clinical classification of disease severity of DR [15]. Sight-threatening diabetic retinopathy (STDR) was defined as the presence of severe, non-proliferative DR, proliferative DR, or diabetic macular edema.

### 2.7. Statistical Analysis

Data was entered, coded, and analyzed using the SPSS version 22.0 statistical program. For descriptive statistics we used the mean, standard deviation, the 95% confidence interval for the mean, and the maximum and minimum values in the case of quantitative variables. In the case of qualitative variables in the studies, such as the prevalence of DR, the statistics used were the absolute and relative frequencies and the percentages of each category.

The meta-analysis to determine heterogeneity and bias between studies was carried out with MetaXL 4.0. [16] that employs the same meta-analysis methods that can be accessed in general statistical packages (such as Stata™) and in dedicated meta-analysis software but makes two additional methods available: the inverse variance heterogeneity and quality effects models. We used the double arcsine transformation method to stabilize the variance of prevalence and used the inverse of the variance of the transformed prevalence as the study weight. For the calculations of the pooled prevalence, cumulative incidence (random effect model), and heterogeneity statistic, Cochran’s Q and the forest plot test were analyzed in the results of the cross-sectional and longitudinal studies.

## 3. Results

### 3.1. Selection of Articles and Documents

Figure 1 is a flowchart with the selection of articles and documents included in the review. At the end of the process, 38 articles, doctoral theses, and documents that met the inclusion and exclusion criteria were selected. For the cross-sectional study we used nine articles and a thesis. For the longitudinal study we used five articles and a doctoral thesis. We used sixteen articles for descriptions of epidemiology and screening of diabetic retinopathy. Finally, we included seven documents that had been published by the MoH and RHCs.

### 3.2. Study of DR Prevalence

Table 1 describes the cross-sectional studies in which the prevalence of diabetic retinopathy in Spain was determined.

The first two studies are those of Santos et al. [17,18] who reported a prevalence of DR of 35.7% in the first and 29.8% in the second. Both were carried out at the RHCS of Extremadura, and differences observed might be due to statistical bias caused by the smaller sample size in the second and to differences in the metabolic control of the patients. Two studies carried out in Catalonia, Teruel et al. [19] and Romero et al. [20], yielded similar prevalence figures, 30.3% in the former and 26.11% in the latter, while two studies from 2009 reported lower prevalence values for DR, 17% for Martínez Rubio et al. [21] and 12.05% for Rodríguez-Villas et al. [22]. López et al. [23] from 2015 found a prevalence of DR of 14.9%, which was based on a sample of 14,266 patients selected from the HOPE cross-sectional study, carried out with T2DM patients from all over Spain between 2009 and 2011. In 2015, Castillo et al. [24] found a prevalence of DR of 8.56%. An article from Rufas et al. [25] was a thesis that gave cross-sectional results at a prevalence of 15.9% and had also performed a five-year study whose incidence results are given in the longitudinal studies section. Finally, Valpuesta et al. [26] found a prevalence of DR of 15%.

### 3.3. Statistical Analysis of Prevalence Studies

The mean prevalence of DR for all studies is 19.93% (95% CI 14.09–27.14, minimum 8.56%–maximum 35.70%) and was 3.08% (minimum 1.84%-maximum 5.30%) from the sight-threatening diabetic retinopathy (STDR) study. The study of the risk factors indicates that DM duration and percentage of DM insulin-treated patients is higher in the studies published before 2008.

Table 1 shows that DM duration was between 12.42 to 14.54 years in the studies carried out before the year 2008 and between 9 to 11.7 years in those carried out after 2008. 

Additionally, between 29% to 34.01% of patients were treated with insulin in studies before 2008; that decrease to 9–11.7% after 2008. There is only one exception from Rufas et al. [25] with 40.8% patients with insulin alone or insulin plus oral hypoglycemic treatment, but that is an exception because many patients were of hospital origin. These two factors could explain the decrease in DR prevalence in studies published after 2008.

Regarding HbA1c levels, there are not enough data in the studies reviewed to be able to perform a statistical meta-analysis.

Figure 2 is the forest plot graph of the meta-analysis by the X MetaXL program.

Figure 2A shows how the studies with prevalences below the mean are located on the left and those with higher prevalences are located on the right. Therefore, we have decided to classify the studies into two groups, before and after 2008. 

The mean prevalence for Group 1 was 28.85% (95% CI 23.14–31.71, minimum 26.11%–maximum 35.7%) and for Group 2 it was 15.28% (95% CI 10.50–22.35, minimum 8.56%–maximum 17.9%).

The statistical analysis of the heterogeneity is high with Q = 1026.26, *p* < 0.001, and I2 = 99% (Figure 2A). 

For each of the two groups, the results are shown in Figure 2B,C. The first group (studies made before 2008) shows a decrease of heterogeneity to Q = 98.68, *p* < 0.001, I2 = 97% (Figure 2B), and Group 2 (studies made after 2008) was Q = 37.83, *p* < 0.001, I2 = 87%.

Figure 2B shows that studies far from the mean were that of Santos et al. from the year 2001 with a prevalence of 35.7% and that of Romero et al. from the year 2008 with a prevalence of 26.11%. 

In group 1, the weight of each study is very similar (right column in Figure 2B); that is, despite the heterogeneity, they all have the same influence. The explanation of heterogeneity in group 1 is due to the studies of Santos et al. [17] made in 2001 to the right of the middle because they have the higher DR prevalence (35.7%) and the Romero et al. [21] made in 2008 and located to the left of the middle with a lower DR prevalence (26.1%), but these data reaffirm that DR prevalence decreases in time.

Figure 2C shows the results of group 2. In group 2, the weight of the analysis is more important for the studies located to the right of the midline. Those of Martinez -Rubio et al. [21] have a 17.9% of DR prevalence. Those of Lopez et al. [23] have a 14.9% DR prevalence. Those of Rufas et al. [25] have a 15.9% DR prevalence. All three studies have a weight higher than 20%. 

The study carried out by Castillo et al. [25] which gives a prevalence of DR of 8.7%, has a weight 13% lower than the aforementioned studies. A possible explanation is that the study was carried out with a small sample size (442 patients) with HbA1c values of 6.92 ± 0.98%, a mean age > 70 years, and 12.4% of patients; all these data suggest that patients had a lower risk of developing DR.

In summary, we observed a decrease in the prevalence of DR over the years.

For STRD, although we only have data from seven of the ten studies, there is a significant difference between the prevalence before the year 2008 with a value of 3.67% (CI95% 1.63–5.13, minimum 1.98%-maximum 5.3%) and after with a value of 1.92% (CI95% 0.82–4.32, minimum 1.84%-maximum 2.29%). We have not performed the meta-analysis as we have data of only seven studies.

### 3.4. Relation between DR Prevalence and DM Diagnosis

In recent years, family physicians made a great effort to diagnose DM early. This has led to an increase in the number of patients known to have T2DM from 2001 to 2017 (see Figure 3). The percentage of patients with DM rises from 5.62% in 2001 to 7.8% in 2017, representing an increase of 27.94% [27].

The increased detection of patients with DM is due to different variables, such as the involvement of general practitioners in the screening of DM [28,29,30,31,32,33]. Likewise, they have been included in DR screening as is the case in other countries such as the U.K. [34,35] using a similar screening circuit with the inclusion of technicians in performing retinographies and general practitioners in DR detection with the support of ophthalmology consultants [36,37,38,39,40]. Additionally, there has been an effort to implement screening programs for DR. The prerequisites conform to the requirements of the British Diabetic Association (BDA) [34]. The protocol begins with the retinography, which is carried out at the primary care center by nursing staff or optometrists trained in obtaining retinal images. The retinographs are filed with the patient’s medical history from which the general practitioner will interpret the images with the support of a consultant ophthalmologist [37,38,39,40,41].

Additionally, Figure 3 shows a decrease in DR prevalence that coincides with the increase in patients diagnosed with DM, and as we can see in the figure, this happens especially from the year 2008. We can deduce that the increase in the detection of patients with DM, many of them in the initial phases, far exceeds the increase in the detection of patients with DR.

### 3.5. Results with Longitudinal Studies of the Application of DR Screening Programs

Table 2 describes the six longitudinal studies that we reviewed. Rodriguez-Acuña et al. [28] with a 10-year follow-up reported a cumulative incidence of 12.2% with a mean annual incidence of 4.45%. The second is Salinero et al. [42], who reported a cumulative incidence of 8.1% at five years with an annual incidence of 2.01%. Rodriguez-Poncelas [43] reported a cumulative incidence of DR of 12.2% at five years with an annual incidence of 2.47%. Rufas et al. [25] reported a cumulative incidence of 15.9% at five years and an annual incidence of 2.2%. Pareja-Ríos et al. [44] reported a cumulative incidence of 16% at eight years and an annual incidence of 6.89%. Finally, Romero-Aroca et al. [45] reported a cumulative incidence of 16% at eight years with an annual incidence of 4.43%.

Figure 4 show the meta-analyses of incidence. We calculated the pooled incidence (random effect model) and heterogeneity statistic Cochran’s Q. Heterogeneity was Q = 4584.85 at a significance of *p* < 0.001 and I2 = 100%. Heterogeneity was high because we mixed three groups of studies with an accumulated incidence at different years.

In Figure 4B,C we observed a decrease in heterogeneity because we applied the analysis according to cumulative incidence at five and eight years, respectively. For studies at five years (Figure 4B), Q = 139.53, *p* < 0.001, and I2 = 99%, and for studies at eight years (Figure 4C), Q = 102.63, *p* < 0.001, and I2 = 98%. The decrease in both studies corresponds to the correct classification of the studies according to the years of follow-up. In any case, the differences in the cumulative incidences in each of the five studies analyzed continue to be important. Surely, the differences may be due to defects in patient follow-ups or the inclusion of new cases of patients with new DM diagnoses.

An interesting fact provided by four of the six studies is that the interval between patient visits, with a mean of 2.72 ± 0.17 years between visits, that does not follow the recommendations of the scientific societies is greater than two years. 

Finally, the annual incidence has been calculated based on the data extracted from the studies or from direct communication with the authors, considering the number of patients for whom DR is diagnosed with respect to the patients screened per year. The average annual incidence of DR found is 3.83% (CI 95% 1.93–5.73, minimum 2.01%—maximum 6.89%).

The value of STDR incidence was 0.41% (CI 95% 0.27–0.55, minimum 0.39%—maximum 0.45%). However, the annual incidence of STR has been calculated based on only three of the six studies; therefore, we have not performed the meta-analysis.

## 4. Discussion

The objective of this review, therefore, was to summarize the current status of DR epidemiology in Spain that depends on the efficacy of DR and DM screening. Because DR screening in Spain is carried out by the RHCs, there are no global data on the epidemiology of DR in Spain. Another added difficulty is that a number of RHCSs implemented them and have published their results, and others have also implemented screening programs but have not so far published any results. Furthermore, there is a lack of a DM patient register which has made our patient sample opportunistic rather than systematic.

Regarding the screening programs established in Spain, the protocol is similar to the U.K. DR-screening program [46,47,48,49] in that with both retinography is caried out by optometrists or ophthalmology nurses, and the image is read by higher-rank professionals who may be primary care GPs. The implication of GPs in the screening seems to be highly effective. It may be one of the causes of the changes observed in the prevalence of DM and the prevalence and incidence of DR. A recent publication [50] reported regular screening of 80% to 100% of patients in the U.K. progam, as achieved by our study, with 91.3% in the Andalusian program, and 61.3% in the Canarias programs [28,44].

In addition, the screening system in Spain is opportunistic and not systematic since there is no unified database of DM patients. This can make it difficult to compare our results with those obtained in other countries that have database systems and centralized data and that allow systematic patient reviews [51,52,53,54,55].

Regarding the prevalence of DR, we reviewed ten publications from 2001 to 2020. The statistical analysis of those studies shows that the prevalence of DR has been decreasing over the years. Since 2008, the prevalence of DR has decreased from 28.85% to 15.28%. The cause of this decrease is linked to a greater diagnosis of patients with DM, especially from the year 2008; thus, in Figure 3 we observe an increase of 27.94% of patients with DM from 2001 to 2017. In addition, patients in the studies conducted after 2008 have a shorter duration of DM and a lower number of patients treated with insulin (Table 1), both important risk factors in the development of DR.

Comparing our study with others published, we must consider that of Li et al. [56], a systematic review of DR for southern European countries (Portugal, Spain, and Italy) with a reported prevalence of 26.5% and data extracted from 13 studies carried out between 1996 and 2009, which is similar to the results of our own studies carried out before 2008.

In addition, their data for DR prevalence was 25.7% across Europe with data extracted from 43 studies carried out between 1995 and 2010 [56]. In other global revision Thomas et al [57] reported a prevalence of 20.6% in Europe for the period 2013 to 2015 similar to our results.

Longitudinal studies published in Spain report the annual incidence determined in the present study was 3.83%, inferior to the 4.6% determined by Li et al. [56] in the U.K. but similar to Dutra Medeiros [58] in Portugal which reports an annual incidence of 3.87% at the three-year follow-up. It is also higher than the incidence of 2.9% obtained by Cheyne et al. [59] in Liverpool in the period of 2013 to 2017.

Although the data on STDR indicates a prevalence of 1.92% and an incidence of 0.41%, it cannot be considered completely accurate given the differences in the definition of STDR between the revised studies. Nevertheless, we can compare our results with those obtained by Nevill et al. [60] in Southwest England during 2013–2016 with an incidence of STDR decreasing from 0.57 in 2013 to 0.35 in 2016. The data are in agreement with those of obtained by Cheyne et al. [59] in Liverpool during 2006 to 2010 with an STDR incidence lower than 2%.

The weaknesses of the current study include the heterogeneity shown between the different published studies and that may be due to differences in data collection as well as the difference in the sample size. Other weaknesses are that the study was carried out with data from only six RHCs which represent 59.61% of the Spanish population; however, since the Spanish population is homogeneous in terms of demographic factors, such as race, age, or gender in the different regions, our results can be extrapolated to the entire Spanish population with diabetes. Additionally, although we believe the data on DR to be reliable, we are not sure about the data on STDR given the differences in diagnosis of the term STDR according to published studies. Finally, in the current study we have not included the study of patients with type 1 diabetes mellitus since these patients are controlled in health service hospitals and, therefore, are not included in DR-screening programs.

## 5. Conclusions

In Spain, from the systematic review via meta-analysis that we carried out, we can calculate that the prevalence value of DR in T2DM patients is currently 15.28% (minimum of 8.59% and maximum of 17.9%), and the STDR prevalence is 1.92% (minimum 1.84% and maximum of 2.29%). The annual DR incidence is 3.83% (minimum of 2.01% and maximum of 6.89%), and the STDR incidence is 0.41% (minimum 0.39% and maximum 0.45%).

## Figures and Tables

**Figure 1 healthcare-10-01318-f001:**
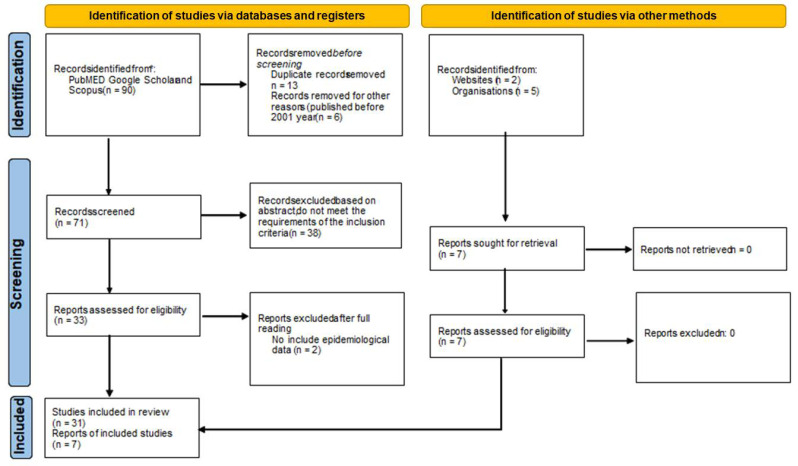
Flow diagram describing the search for publications on screening programs and results in Spain.

**Figure 2 healthcare-10-01318-f002:**
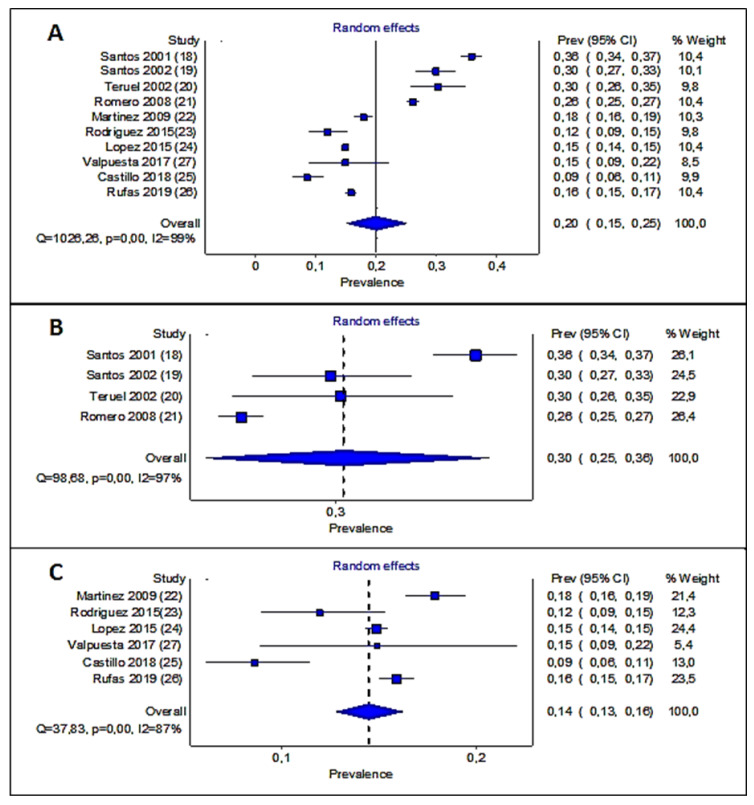
Forest plot graph of the meta-analysis has been constructed taking into account that the average prevalence is 19.93%. (**A**) data of all studies, (**B**) studies conducted between 2001 to 2008, (**C**) studies conducted between 2009 to 2019.

**Figure 3 healthcare-10-01318-f003:**
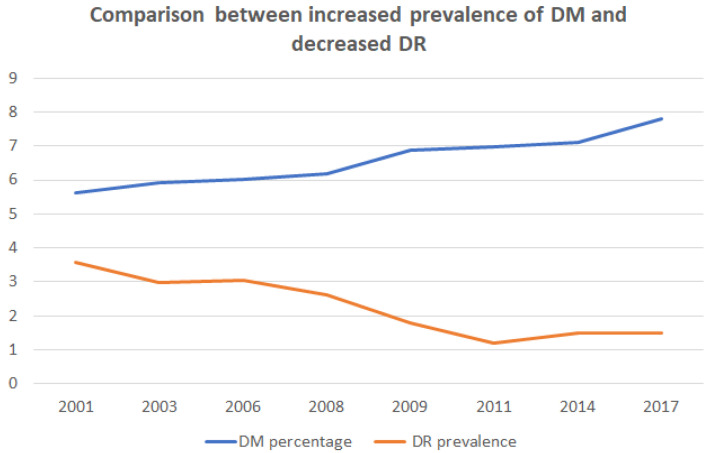
Relationship between the increase in the prevalence of diabetes mellitus versus the decrease in the prevalence of diabetic retinopathy.

**Figure 4 healthcare-10-01318-f004:**
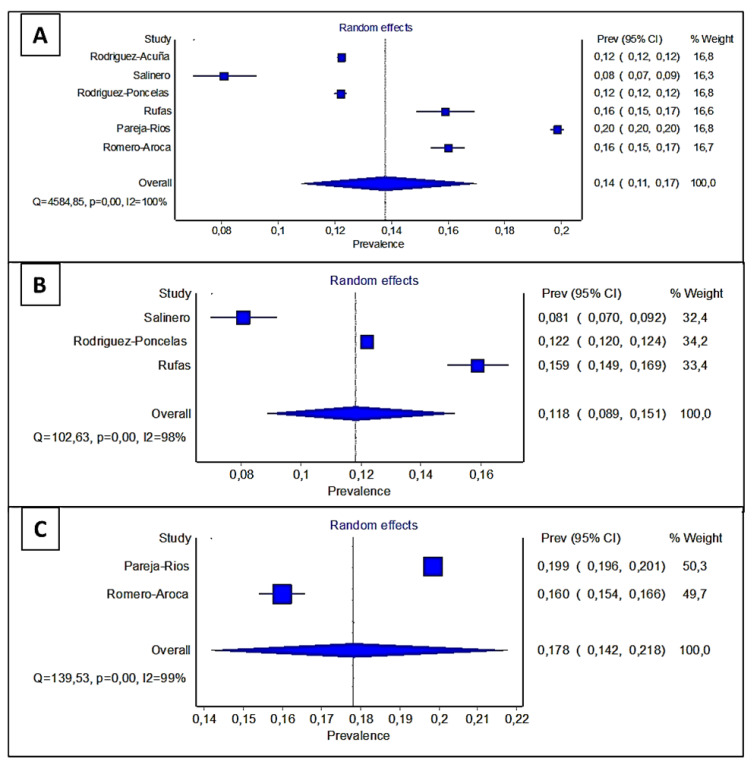
Forest plot of the reviewed publications on incidence. (**A**) Data of all studies, (**B**) data of studies with 5 years follow-up, (**C**) data of studies with 8 years follow-up.

**Table 1 healthcare-10-01318-t001:** Diabetic retinopathy outcomes and demographic data from cross-sectional studies.

Autor	Santos et al.		Santos et al.	Teruel et al.	Romero et al.	Martinez-Rubio M	Rodriguez-Villas et al.	Lopez et al.	Castillo et al.	Rufas et al.	Valpuesta et al.
Year *	2001		2002	2002	2008	2009	2015	2015	2015	2015	2019
Patients with DR/ Total patients	1112/3114		226/762	121/401	2123/8187	436/2435	47/394	2126/	38/442	1001/6294	17/114
Patients with STDR/Total patients	165/3114		37/762	10/401	162/8187	56/2435	ND	ND	8/442	ND	2/114
Mean age	63.2 ± 13.4		66.2 ± 11.4	ND	64.6 ± 10.78	ND	70.4 ± 11.64	64.3 ± 11.2	70 ± 10.6	70.3	68.69 ± 9.85
Duration of DM in years	12.75 ± 6.7		13.81 ± 5.4	14.54 ± 7.2	12.42 ± 6.3	ND	9.02 ± 2.08	9.0 ± 7,1	11.7 ± 7	11	9.8 ± 8.43
M/W %	37.6/62.4		39.2/60.8	49/51	55.21/44.79	56.46/43.54	57.9/42.1	48.1/51.9	55.9/44.1	56/44	58.8/41.2
DM treatment						ND					
Diet	19.4		18.2	22	16.94		6.1		12.4	8.1	5.5
Oral	49.5		52.82	51.99	51.12		73.8		67.3	51.5	76.3
IT / IT± oral	31.1		29	32.01	31.54		20.1		20.3	40.8	18.4
Arterial hypertension	47.2		36	49	68.36	ND	71	74.1	78.8	69	ND
HbA1c	ND		ND	ND	7.34 ± 1.23%	ND	7.23 ± 1.34%	7.30%	6.92 ± 0.98%	6.8	ND
Diabetic retinopathy											
DR prevalence	35.70%		29.80%	30.30%	26.11%	17.90%	12.05%	14.9	8.56%	15.90%	15%
STDR prevalence	5.30%		4.80%	2.50%	1.98%	2.29%	ND	ND	1.81%	ND	1.74%.

Abbreviations: year * = year of realization, DM = diabetes mellitus, M/W = man/women, diet = DM treatment with diet, oral = treatment with oral hypoglycemic drugs, IT = insulin treatment ± oral hypoglycemic drugs, DR= diabetic retinopathy, STDR = sight-threatening diabetic retinopathy, ND = no data available.

**Table 2 healthcare-10-01318-t002:** Diabetic Retinopathy Outcomes and Demographics from Longitudinal Studies.

	10-Year Follow-Up	5-Year Follow-Up			8-Year Follow-Up	
Author	Rodriguez-Acuña et al.	Salinero et al.	Rodriguez-Poncelas et al.	Rufas et al.	Pareja-Ríos et al.	Romero-Aroca et al.
Study dates *	2008–2018	2007–2011	2008–2012	2010–2015	2011–2019	2008–2015
Patients with DR / Total patients	/	194/2405	/	808/4276	/	2462/
DM duration at baseline (years)	6.4 ± 6.9	7.7	7.6 ± 5.6	11	ND **	7.37 ± 6.92
Women/Men at baseline (%)	54.6/45.4	39.2/60.8	43.8/56.2	56/44	ND	42.7/57.3
Age at baseline (years)	62.8 ± 12.8	67.5 ± 10.6	66.91 ± 11	70.3	ND	64.74 ± 12.39
Diabetic retinopathy						
Interval between visits	2.9	ND	ND	2.8	2.7	2.5
Cumulative incidence ***	12.2% at 10 years	8.1% at 5 years	12.2% at 5 years	15.9% at 5 years	19.9% at 8 years	16% at 8 years
Annual incidence of DR ****	4.45%	2.01%	2.47%	3.2%	6.89	4.43%
Annual incidence of STDR	0.45%	ND	0.35%	ND	ND	0.44%

Footnotes * The data has been extracted from each of the published studies or through direct communication with the authors. ** No data. *** The cumulative incidence has been calculated based on the number of patients with DR with respect to the total number of patients in the sample. **** The annual incidence has been calculated based on the cumulative incidence or if the authors of each study included it in their publication.

## Data Availability

The data is available in the link www.iispv.cat (accessed on 10 July 2022)/Home—IISPV.

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
