# Peer review of "Changes in the Epidemiology of Diabetic Retinopathy in Spain: A Systematic Review and Meta-Analysis"

_healthcare, 2022, doi:10.3390/healthcare10071318_

Round 1

Reviewer 1 Report

This is a well written review article that will contribute to the literature. 

Author Response

Question. This is a well written review article that will contribute to the literature

Response. Thank you very much for the reviewer's comments and for the time you have spent reviewing this manuscript.

Reviewer 2 Report

This paper has a potential to be accepted after minor revision.

The search strategies for all included databases should be  provided in the supplementary materials...

The discussion part should contain more comparative references.

Author Response

Commentary. This paper has a potential to be accepted after minor revision.

Question 1. The search strategies for all included databases should be provided in the supplementary materials...

Response to question 1. Thank you very much for your comment. Authors have included supplementary material with the description of the databases.

Question 2. The discussion part should contain more comparative references.

Response to question 2. Thank you very much for your comments. Authors have included more references in the discussion.

Reviewer 3 Report

The article is well written. Just note that other studies (prospective or retrospective) are better to answer your research question. Please refer to the PRISMA guideline for reporting in systematic reviews and meta-analyses.

Author Response

Commentary. The article is well written. Just note that other studies (prospective or retrospective) are better to answer your research question. Please refer to the PRISMA guideline for reporting in systematic reviews and meta-analyses.

Response. Thank you very much for your insightful comments.

Authors ae totally agree with your statement that other prospective or retrospective studies that are carried out at the population level throughout Spain based on indications from the Spanish Ministry of Health can give excellent results, but currently, as health is transferred to the Different autonomous communities of Spain, there is a difficulty in being able to carry out this type of study, the authors hope that this manuscript will be a revulsive that makes epidemiological studies at the national level in Spain possible.  

The authors have taken into account the PRISMA guide to prepare the present study, reviewing all the published articles that referred to the epidemiology of diabetic retinopathy in Spain. We have applied the norms indicated in PRISMA for review articles with meta-analysis to obtain the results. In methods I included the paragraph, “Following the guideline of Preferred Reporting Items for Systematic Reviews and Me-ta-Analyses (PRISMA) [11] and the nternational Guidelines for Systematic Literature Reviews and Meta-Analyses of Observational Studies and Epidemiology (MOOSE) [12].”

Reviewer 4 Report

This manuscript is a systematic review integrating previous epidemiologic studies conducted in Spain for diabetic retinopathy.

The manuscript was well prepared. Moreover, the cited and their discussion are presented in good style. However, authors have some minor points that needs to be addressed.

・Please unify significant figures, en-dashes, em-dashes, commas, decimal points, etc. throughout the manuscript.

・Please correct the arrows in Figure 1 to be straight.

・The decimal point in Figure 4 is a comma, which is difficult to read.

・Please replaces the low resolution and quality of the some figure with cleaner figures.

I hope this manuscript will be published more quickly.

Author Response

Commentary. This manuscript is a systematic review integrating previous epidemiologic studies conducted in Spain for diabetic retinopathy.

The manuscript was well prepared. Moreover, the cited and their discussion are presented in good style. However, authors have some minor points that needs to be addressed.

Question 1. Please unify significant figures, en-dashes, em-dashes, commas, decimal points, etc. throughout the manuscript.

Response. Thank you very much for your comment. We have replaced the commas by points in the figures that required it as well as in the text.

Question 2. Please correct the arrows in Figure 1 to be straight.

Response We have corrected the arrows in figure 1.

Question 3. The decimal point in Figure 4 is a comma, which is difficult to

read.

Response. We have corrected the decimal point in figure 4

Question 4. Please replaces the low resolution and quality of some figure with cleaner figures.

Response. We have improved the quality of the figures with TIFF.

Reviewer 5 Report

Thanks for inviting me t review this paper that described the trends of diabetic retinopathy prevalence in Spain. Generally, the manuscript is well written. I found that the prevalence of DM raised since 2001 and that of diabetic retinopathy decreased in the meanwhile. Is it possible that primary physicians are more likely to diagnose DM at an earlier stage so that the prevalence of diabetic retinopathy decreases because of a larger denominator? Would the authors commend on this?

Author Response

Commentary. Thanks for inviting me to review this paper that described the trends of diabetic retinopathy prevalence in Spain. Generally, the manuscript is well written. I found that the prevalence of DM raised since 2001 and that of diabetic retinopathy decreased in the meanwhile. Is it possible that primary physicians are more likely to diagnose DM at an earlier stage so that the prevalence of diabetic retinopathy decreases because of a larger denominator? Would the authors commend on this?

Response to reviewer 5. Thank you very much for reviewing the manuscript.

Regarding the question, in the manuscript we explain that as of 2006, the diagnosis of patients with diabetes mellitus who present forms of recent diagnosis increases, that is, the number of diabetic patients with a short duration of DM and who are treated with diet or with oral antidiabetics, either have a lower risk of developing diabetic retinopathy.

On the other hand, it is true that the participation of general practitioners in the control of diabetes mellitus increases, increasing the number of patients diagnosed with diabetes mellitus.

Round 2

Reviewer 3 Report

To me, the design of this study is not convincing/correct. The design has problems/concerns and biases that cannot be corrected. I recommend that a methodologist/epidemiologist criticize/review it.